# Self-Relevance Moderates the Relationship between Depressive Symptoms and Corrugator Activity during the Imagination of Personal Episodic Events

**DOI:** 10.3390/brainsci13060843

**Published:** 2023-05-23

**Authors:** Leonard Faul, Jane M. Rothrock, Kevin S. LaBar

**Affiliations:** Center for Cognitive Neuroscience, Duke University, Durham, NC 27708, USA; leonard.faul@duke.edu (L.F.); jane.rothrock@duke.edu (J.M.R.)

**Keywords:** emotion context insensitivity, depression, self-relevance, imagination, corrugator, electromyography, autobiographical memory

## Abstract

Accumulating evidence suggests depression is associated with blunted reactivity to positive and negative stimuli, known as emotion context insensitivity (ECI). However, ECI is not consistently observed in the literature, suggesting moderators that influence its presence. We propose self-relevance as one such moderator, with ECI most apparent when self-relevance is low. We examined this proposal by measuring self-report and facial electromyography (EMG) from the corrugator muscle while participants (n = 81) imagined hypothetical scenarios with varying self-relevance and recalled autobiographical memories. Increased depressive symptoms on the Center for Epidemiologic Studies Depression Scale were associated with less differentiated arousal and self-relevance ratings between happy, neutral, and sad scenarios. EMG analyses further revealed that individuals with high depressive symptoms exhibited blunted corrugator reactivity (reduced differentiation) for sad, neutral, and happy scenarios with low self-relevance, while corrugator reactivity remained sensitive to valence for highly self-relevant scenarios. By comparison, in individuals with low depressive symptoms, corrugator activity differentiated valence regardless of stimulus self-relevance. Supporting a role for self-relevance in shaping ECI, we observed no depression-related differences in emotional reactivity when participants recalled highly self-relevant happy or sad autobiographical memories. Our findings suggest ECI is primarily associated with blunted reactivity towards material deemed low in self-relevance.

## 1. Introduction

How we respond to emotional events is subject to a myriad of internal and external influences that collectively shape both subjective and physiological indices of affect. Of particular importance is one’s generalized mood state, which alters the neural and physiological correlates of experienced affect [1,2,3], produces mood-congruent appraisals of emotional events [4], and shifts attention to be more self-focused when we are in negative moods [5]. Accordingly, within the clinical domain, decades of research have shown that major depressive disorder is associated with aberrant patterns of emotional responding compared to healthy controls [6]. Yet, despite this body of work, our understanding of emotional reactivity patterns in depression remains relatively imprecise. Diagnostic criteria and treatments for depression predominantly focus on alterations in mood state, while placing comparatively less emphasis on alterations in emotional reactivity due to a lack of specificity in our understanding of such effects [7].

Indeed, empirical work on this topic has produced mixed findings for the relationship between depressive symptoms and emotional response, often supporting one of three dominant views in the field [7]. The first view suggests that depression is primarily reflected in blunted reactivity to positive information, otherwise referred to as positive attenuation, which is thought to be associated with reduced striatal dopamine functioning [8]. Accordingly, on reward-based paradigms, depressed individuals exhibit a general insensitivity to and lack of engagement with rewarding, positive content that is reflected in diminished corticostriatal engagement [9,10]. However, others have found that the presence of positive attenuation may be sensitive to the self-relevance and ecological validity of experimental assessments. Recent research with ecological momentary assessments (EMA), for instance, have shown that depressed individuals sometimes respond to real-life positive events with greater reactivity than healthy controls [11,12,13]. This amplified response profile in subjective reports has been theorized to reflect low expectations for (and less occurrences of) positive experiences that contrast with one’s depressed mood, thus facilitating heightened reactivity when such events occur [12]. Positive reactivity is therefore not universally attenuated in depression, and ongoing research is seeking to reconcile these effects with the proposal that self-relevance and ecological validity of experienced events may play a moderating role [6,14].

Another view posits that depression is primarily characterized by negative potentiation, such that pervasive depressed mood and negative self-schemas facilitate stronger reactions to negative content. This proposal aligns with cognitive models of depression that center on negative thoughts about the self, world, and future [15]. Indeed, depression is associated with difficulties disengaging from negative material, greater endorsement of negative self-schemas, and biased recall for negative experiences, especially when such content is personally relevant [16,17,18,19,20]. Surprisingly, though, research on emotional expression has been equivocal, with an accumulating body of work indicating reduced emotional reactivity in depression to negative stimuli [6,7,14,21]. That is, even though depression is associated with negative biases in attention and memory, empirical evidence suggests that initial emotional reactivity to negative content is blunted.

Consistent observations of blunted emotional reactivity in depression towards both positive and negative material has brought about a third account for emotional responding known as emotion context insensitivity (ECI), which suggests that depression is associated with a generalized loss of reactivity to all emotional stimuli. ECI is supported by meta-analytic evidence for flattened emotional reactivity in depression regardless of stimulus valence [7]. ECI also converges with research on emotional inertia that indicates depression is associated with greater resistance to emotional change [22] and that depressed individuals exhibit less differentiation between discrete negative affective states [23]. Indeed, these converging findings suggest that ECI may be thought of as reflecting a general loss of dissociation between affective states [24]. According to evolutionary perspectives, the presence of ECI in depression is explained by mood-associated reductions in motivated action and increases in internal cognitive focus, given that continued action in the context of loss may be wasteful or even dangerous [6,14,25]. For instance, diminished emotional reactivity and reduced motivated behavior facilitate disengaging from unreachable goals [25]. Increasing behavioral and physiological evidence for ECI over the years has led to this view becoming a dominant perspective in the field [6,14].

Despite this empirical support, ECI is not always observed in the literature, and recent reviews have acknowledged a lack of understanding for the boundary conditions of this effect [6,14]. As mentioned, depression has been shown to associate with enhanced reactivity to real-life positive events [11,12,13], and the emotional blunting effects of ECI have been difficult to integrate with negative biases in attention and memory that are prevalent in depression [16]. More work is needed to understand which conditions predominantly facilitate ECI in depressed individuals, as well as whether ECI generalizes across all indices of emotional reactivity or is more specific to some measures than others.

Accordingly, here we propose two key moderators that determine the presence and/or strength of ECI. First, we suggest that ECI diminishes as emotional material becomes more personally relevant. This hypothesis is informed by multiple literatures, including the amplified positive reactivity effects seen in EMA studies of personally experienced events [11,12,13], as well as empirical evidence indicating that increased self-relevance bolsters the presence of negative attention and memory biases in depressed mood [18]. Moreover, this proposal aligns with findings of enhanced self-focused attention in depression [26,27], as well as recent demonstrations of increased neural sensitivity (via electrophysiology recordings) to self-relevant stimuli in major depressive disorder [28]. We suggest that depression may not be associated with reduced emotional reactivity in all instances, but rather that depression makes emotional reactivity more selective towards emotional stimuli that are more personally relevant. Depression may facilitate blunted reactivity towards less self-relevant stimuli but retained or even enhanced reactivity towards highly self-relevant stimuli. This sensitivity may be difficult for researchers to appreciate if using stimulus materials that generally lack personal relevance and/or if experimental tasks do not provide participants the opportunity to rate the personal relevance of each stimulus.

We also propose that the chosen index of emotional reactivity shapes the extent of ECI. That is, even when researchers use personally relevant stimuli or acquire such ratings, patterns of emotional reactivity in depression may differentially manifest depending on how reactivity is measured. While emotional response coherence across behavioral and physiological measures has been shown for both positive and negative affect, variability in the degree of this coherence suggests important individual differences that need to be studied [29]. Recent work has shown, for instance, that lower reports of emotional well-being are associated with weaker coherence between physiological and subjective measures of emotion [30]. Increased depressive symptoms may therefore be differentially linked with ECI depending on what is measured due to the presence of reduced emotional response coherence. Subjective reports and physiological measures offer complimentary information about an emotional response, and it is prudent to gather both when trying to accurately assess emotional response profiles.

Therefore, to test these proposed moderators, here we analyzed both self-report and physiological indices of emotional reactivity while participants with varying degrees of depressive symptoms were exposed to self-relevant emotional stimuli. Specifically, our stimulus set consisted of brief hypothetical scenarios written in the second-person perspective, and we facilitated self-referential emotional responses by asking participants to imagine themselves within the depicted events. Use of mental imagery in the laboratory has been shown to activate similar behavioral and physiological response profiles as reactions to real experiences [31,32,33,34]. Thus, due to their self-referential nature, these hypothetical scenarios might evoke reactions that better reflect how people respond to actual events in daily life. Moreover, after imagining these scenarios, we also asked the same participants to recall either happy or sad autobiographical memories (between groups) to examine the presence or absence of ECI for reactions to personal episodic events that were actually experienced. For our measure of physiological response, we focused on facial electromyography (EMG) from the corrugator muscle, which has been shown to be positively correlated with reports of negative affect [35] and linked with neural response in the medial prefrontal cortex and amygdala—areas considered instrumental in neurocognitive models of depression [36]. We chose EMG as our measure of physiology given its specificity in dissociating emotional responses on the basis of valence (and thus suitable for testing ECI), whereas other physiological measures such as skin conductance are more sensitive to arousal and thus less useful in dissociating between responses to positive and negative stimuli [37]. We hypothesized that EMG corrugator amplitude would exhibit depression-associated ECI, such that increased depressive symptoms would be associated with blunted corrugator reactivity when imagining emotional scenarios. Importantly, though, we hypothesized that this blunted reactivity would be moderated by the self-relevance of the depicted events, whereby ECI would be most apparent for scenarios low in self-relevance but less apparent as self-relevance increases.

## 2. Materials and Methods

### 2.1. Participants and Procedure

The current investigation is part of a larger multi-session study on mood-congruent memory that consisted of three sessions: Session 1 (online) where participants provided happy and sad autobiographical memory cues, Session 2 (lab—encoding) where participants imagined hypothetical scenarios and recalled either the happy or sad autobiographical memories (as part of a mood induction), and Session 3 (lab—recall) that consisted of a memory test for the imagined scenarios. Given that the focus of the present investigation is to examine affective experience as it relates to depression, we present here the behavioral and physiological (corrugator) responses that were measured at Session 2 during the imagination and autobiographical memory tasks and their association with individual differences in depressive symptoms. The overall study aimed to collect data from at least 80 individuals in order to have sufficient power to test the influence of mood on next-day memory, and thus recruited a total of 98 participants, 88 of whom successfully completed Session 2, which we will focus on for our analysis of ECI. Inclusion criteria included an age range of 18–39 years old, as well as no history of a neurological condition or psychiatric disorder (although we did not specifically test for the presence or absence of a disorder).

For the present analyses, we further excluded outlier participants (>3 SD above or below the mean) based on performance on the baseline even/odd active fixation task (see Section 2.2) and state mood reports. Specifically, we excluded two participants with poor accuracy on the even/odd task, as these participants were generally inattentive during the scenario imagination task. We further excluded two participants who self-reported abnormally high levels of mood disturbance immediately before or after the imagination task, as well as one participant who reported high levels of mood disturbance after recalling happy memories during the autobiographical recall task. Finally, we excluded two participants for significant loss of EMG data during the imagination task (>20% of data) after applying preprocessing and cleaning procedures (see Section 2.5). Applying these exclusionary criteria resulted in a final sample of 81 participants (mean age = 21.3 years, SD = 3.4; 49 female, 32 male; 40 participants in the happy memory recall group and 41 participants in the sad memory recall group). All participants provided written informed consent in accordance with the Duke University Institutional Review Board. Twelve participants received course credit for participating, while the other 69 participants received monetary compensation ($15/h).

Session 1 was completed through Qualtrics, where participants provided consent, answered demographic questions, completed psychosocial questionnaires, and created cue words for their autobiographical memories. Questionnaires included the Center for Epidemiologic Studies Depression Scale, CES-D [38], the Ruminative Response Scale, RRS [39], and the Social Desirability Scale, SDS-17 [40]. We used the CES-D as our primary measure of depression for ECI analyses, which consists of 20 items asking participants to indicate how frequently they have experienced a range of depressive symptoms over the past week, from 0 (rarely or none of the time) to 3 (most or all of the time). Thereafter, participants were asked to provide 15 happy autobiographical memory cues and 15 sad autobiographical memory cues (in alternating order) for events that occurred within the past five years by writing a brief description and title of the memory (to be used as cues at Session 2), the approximate date and general location of the event (to encourage specificity), and ratings of valence (1—very sad to 7—very happy), arousal (1—very calm to 7—very intense), and detail (1—very vague to 7—very clear). Session 1 ended once all memory responses were complete.

Session 2 took place the following week (6–8 days later) in our laboratory space between the hours of 11:00 a.m. and 6:00 p.m. Participants first completed the Subjective Units of Distress Scale (SUDS), which indexes current level of distress anywhere from 0 (totally relaxed) to 10 (highest distress that you have ever felt). We next prepared participants for recording of electromyography from the corrugator supercilii (see Section 2.5 for more details). We ensured that all equipment was working appropriately by having participants complete a brief practice task where they identified numbers as even or odd on the computer screen (note that this even/odd task also separated trials in the imagination task). Participants next completed the imagination task, during which they imagined hypothetical scenarios and provided ratings of valence, arousal, and self-relevance (see Section 2.2 for more details). Thereafter, participants completed a short filler task for approximately five minutes consisting of a basic number search puzzle, then completed the autobiographical recall task, during which they recalled a series of either happy or sad memories (see Section 2.3 for more details). Before finishing the session, participants provided another SUDS rating. Participants (n = 4) who responded above two points from their starting SUDS rating were asked to watch a neutral video of a train traveling through British Columbia (Highball Productions) before completing a final SUDS rating. We implemented this procedure to ensure that participants did not leave the study session while still experiencing high levels of distress after recalling sad memories. We also acquired self-report ratings of mood before the imagination task, after the imagination task, and before the autobiographical recall task using the Profile of Mood States questionnaire [41]. Participants were instructed to respond based on how they feel “right now”. Total mood disturbance scores were calculated by subtracting the sum of all negative subscales from the sum of all positive subscales. As mentioned, these mood ratings were used in the present study to exclude participants who reported abnormally high levels of total mood disturbance during the experimental session. We applied these exclusions to isolate behavioral and physiological reactivity related to more general indices of depressive symptoms, as opposed to momentary negative affective states that were unique to these few outlier participants.

### 2.2. Imagination Task

During the imagination task, participants were instructed to read and imagine brief, hypothetical scenarios that varied in their emotional outcomes. These scenarios were obtained from previous studies that have examined the role of self-relevance in moderating the neural processing of emotional imagined events [42,43,44,45]. Each scenario consisted of two sentences written in the present tense and second-person perspective, with the first sentence introducing a situation with neutral or ambiguous valence and the second sentence continuing the scenario with a specific emotional outcome (e.g., normative sad example: “After some thought, you show up at an event. Everyone is disappointed when you arrive”; normative neutral/ambiguous example: “You are presenting your paper in front of your class. You get a lot of feedback at the end”; normative happy example: “At your job, all the employees get yearly reviews. Your evaluation is quite encouraging this year”). These stimuli have been shown to elicit discrete affective experiences that parallel the emotional space of other modalities (e.g., movies, music, facial expressions, etc.) while also facilitating more self-relevant emotional states [46]. For the present study, we selected 45 scenarios based on normative affect ratings, such that the stimulus set consisted of 15 happy, 15 neutral, and 15 sad scenarios. These normative ratings were acquired from a separate group of online participants recruited via Amazon’s Mechanical Turk (n = 90; 42 female, 48 male; mean age = 34.8 years, SD = 8.5), with each scenario rated by 15 different raters on various affect scales. The happy, neutral, and sad scenarios selected for this study were significantly different from one another in average normative ratings of sadness, happiness, and general valence (all *p*s < 0.01). Happy and sad scenarios did not differ in arousal (*p* = 1), and both were more arousing than neutral scenarios (both *p*s < 0.001).

For each trial in the imagination task, participants were first presented with the two-sentence scenario on the computer screen for a total of 16 s. During the first eight seconds only the scenario text was shown, and participants were instructed to press the space bar on the keyboard once they had finished reading the text and started imagining the scenario. At eight seconds into the imagination period the words “I feel…” appeared below the scenario text to remind participants to focus on how they feel as they imagine the depicted event and prepare for the behavioral ratings. After the sixteen second imagination period, participants were then provided four seconds for each of three ratings (presented in a random order): valence (1—very sad, 4—neutral, 7—very happy), arousal (1—very calm to 7—very intense), and self-relevance (likelihood of the depicted event occurring in real life; 1—very unlikely to 7—very likely). Trials were separated by an active fixation task consisting of labeling a number in the center of the screen as even (pressing the G key) or odd (pressing the H key) within a 2.5 s window, and a passive fixation period of 4–5 s before the onset of the next trial. Prior to the start of the imagination task, participants were provided with three practice trials to familiarize themselves with the timing of all task components.

### 2.3. Autobiographical Recall Task

During the autobiographical recall task, participants were shown either the 15 happy or 15 sad memories they had provided in an online session a week prior. Before starting the task, participants were told that they would only see one set of memories (either happy or sad) but were not explicitly told which set would be seen. The autobiographical recall task mirrored the imagination task such that each trial began with recalling a memory and then providing behavioral ratings of valence and arousal. The memory cue was presented in the center of the screen for a total of 20 s, with the words “I feel…” appearing below the cue after 10 s had passed. Participants were instructed to press the space bar on the keyboard once they had retrieved the event from memory and started reminiscing on the details of the experience. Participants were provided 4 s to make each behavioral rating (presented in a random order), and trials were separated by a passive fixation cross that lasted for 5–6 s. Throughout the recall task, emotion-congruent instrumental music (sad or happy) played via speakers placed next to the computer screen to help facilitate feelings of sadness or happiness. These music clips were selected based on song attributes provided by the Spotify API tool Sort Your Music: sortyourmusic.playlistmachinery.com (accessed on 22 January 2020). On average, the music clips did not significantly differ between the happy and sad inductions on beats per minute (*p* = 0.870), energy/arousal (*p* = 0.582), or duration (*p* = 0.557), but did significantly differ in their valence (*p* < 0.001). The happy instrumental music clips consisted of *The Wheel* by Dan Musselman (3:42), *Love Breaths* by Mark Andrew Hansen (2:26) and *Remembered Sundays* by CatKids (2:47). The sad instrumental music clips consisted of *Lost on the Moon* by Sad Instrumental Piano Music Zone (2:58), *Empty Bed* by Sad Instrumental Piano Music Zone (3:28), and *Pessimistic Thoughts in my Head* by Sad Instrumental Piano Music Zone (3:15).

### 2.4. Behavioral Analysis

All behavioral analyses were performed in R. We used linear mixed-effects regression models fit by maximum likelihood to examine the relations among depressive symptoms and scenario ratings with the lme4 package [47]. Subjects and scenarios were both modeled as random effects. Simple effects/slopes were examined with the emmeans package [48], with Bonferroni-corrected pairwise comparisons. We report all betas as standardized coefficients. Our models took the following structures:Valence ~ Depressive Sym × Normative Category + (1|subject) + (1|scenario)
Arousal ~ Depressive Sym × (Valence + I(Valence^2)) + (1|subject) + (1|scenario)
Self-Relevance ~ Depressive Sym × Valence + (1|subject) + (1|scenario)

### 2.5. EMG Acquisition, Preprocessing, and Analysis

Facial EMG data were collected from the right corrugator muscle using 2 Ag-AgCl reusable shielded electrodes with a 4 mm diameter contact area (BIOPAC Systems; Goleta, CA, USA). The two electrodes were placed above the medial end of the right eyebrow with an adhesive disk and separated horizontally by approximately 1–2 cm from center to center. Prior to placing the electrodes, the skin was cleaned with an alcohol wipe, and we filled the electrodes with GEL100 (BIOPAC Systems; Goleta, CA, USA) to facilitate conductance. The raw EMG signal was sampled at a frequency of 1 kHz using AcqKnowledge software and BIOPAC MP-160 hardware (BIOPAC Systems; Goleta, CA, USA) with a high-pass filter of 10 Hz, a low-pass filter of 500 Hz, a notch filter of 60 Hz, and a gain setting of 5000. Post-collection, EMG data were preprocessed via rectification and smoothing with a 40-Hz low-pass filter using the neurokit2 Python toolbox [49], as well as downsampled to 62.5 Hz. To diminish the impact of any remaining spikes in the data, we used the despike function from the oce package in R [50], specifying a running median window size of 15 samples, a threshold of 3 SD for identifying spikes, and replacing spike values with the referenced running median value.

Preprocessed EMG data from the imagination task were analyzed by first extracting samples from each trial’s baseline and imagination period (−2 to 16 s after scenario onset). Samples were then averaged together into two-second bins. Baseline-corrected EMG amplitude was quantified as the ratio of corrugator amplitude in each bin when compared to the two-second pre-trial baseline (−2 to 0 s). To minimize the impact of any remaining outlier activity (e.g., resulting from motion artifacts), we removed trials in which the average baseline-corrected response across the full imagination period was greater than 3 SD from the mean response for that scenario across participants. For most participants, this procedure resulted in the removal of no trials (n = 56) or only one trial (n = 11). As noted above, two participants were excluded from all analyses for losing greater than 20% of trials at this stage.

As with the behavioral analyses, we used linear mixed-effects regression to examine corrugator activity as a function of time, scenario characteristics, and depressive symptoms, while modeling both subjects and scenarios as random effects. Lower-level interactions (joint tests) and simple effects were examined with the emmeans package. Our primary model of interest took the following structure for the average corrugator response during scenario imagination:Corrugator ~ Dep Sym × (Val + I(Val^2)) × Self-Relevance + (1|subject) + (1|scenario)

We applied a similar approach to examining preprocessed EMG data from the autobiographical recall task by extracting each trial’s baseline and recall period (−2 to 20 s after memory cue onset), averaging samples into two-second time bins, and calculating baseline-corrected EMG amplitude. We removed trials in which the average baseline-corrected response across the full recall period was greater than 3 SD from the mean response for that trial number across participants within the same recall group. As before, most participants retained all trials (n = 63) or lost only one trial (n = 13). One additional participant, however, was removed from this analysis for losing nearly half of their trials.

Again, we used linear mixed-effects regression to examine corrugator activity as a function of time, recall group, and depressive symptoms, while modeling subjects as random effects, but not memories since they differed across participants (whereas scenarios were the same across participants). However, when examining the estimated marginal means of corrugator amplitude across time bins, we also included a random effect of memory nested within subjects to accommodate differences in the time course of corrugator activity depending on the exact memory recalled. We did not model valence ratings, given that the task was designed to only recall happy or sad memories and therefore we did not have the same distribution of valence ratings as with the imagination task. Instead, our primary model of interest took the following structure for the average corrugator response during memory recall:Corrugator ~ Depressive Symptoms × Recall Group + (1|subject)

## 3. Results

To examine patterns of ECI in depression with varying levels of self-relevance, we first analyzed behavioral (Section 3.1) and physiological (Section 3.2) responses to imagined emotional events during the scenario imagination task. We then examined whether similar patterns exist in affective responses during recall of happy or sad autobiographical memories (Section 3.3).

### 3.1. Depression-Associated ECI Is Found in Ratings of Arousal and Self-Relevance but Not Valence

Distribution of self-reported depressive symptoms as measured with the CES-D questionnaire are shown in Figure 1a. The possible range of scores on the CES-D is 0–60, with higher scores indicating the presence of more symptomatology. Scores in the range of 16–20 have been shown to be appropriate cutoff scores for identifying individuals at risk for clinical depression [51]. Participants in the present study reported sum scores within a range of 0–38, with an average of 13.9 (SD = 8.3) and a median value of 12. Examining the distribution of sum scores revealed that 38% of participants (n = 31) scored 0–9, 26% (n = 21) scored 10–15, and 36% (n = 29) scored 16 or above (at risk for depression).

To examine behavioral biases in experienced affect, we regressed depressive symptoms with each rating acquired during the imagination task (valence, arousal, and self-relevance). When scenarios were grouped by normative emotion categories (sad, neutral, and happy), we observed a main effect of emotion category on valence ratings (F_2,45_ = 254.155, *p* < 0.001), such that happy scenarios were rated as more happy than neutral scenarios (*β* = 0.918, *t*_48.1_ = 10.035, *p* < 0.001, 95% CI = [0.691, 1.140]), and neutral scenarios were in turn rated as more happy than sad scenarios (*β* = 1.074, *t*_48.1_ = 11.735, *p* < 0.001, 95% CI = [0.847, 1.300]). Importantly, this main effect was qualified by a significant interaction with depressive symptoms (Figure 1b; F_2,3461.2_ = 11.202, *p* < 0.001). Examination of simple slopes revealed that individuals reporting higher depressive symptoms rated happy scenarios as being more happy (*β* = 0.058, *t*_289_ = 3.225, *p* = 0.001, 95% CI = [0.022, 0.093]) and sad scenarios as being more sad (*β* = −0.043, *t*_291_ = −2.405, *p* = 0.017, 95% CI = [−0.078, −0.008]), with no difference in ratings for neutral scenarios (*β* = 0.009, *t*_294_ = 0.505, *p* = 0.614, 95% CI = [−0.026, 0.044]).

Given these individual differences in the reported valence of the scenarios, all subsequent analyses used ratings of valence provided by participants during the imagination task instead of pre-assigned normative categories. With this approach, we next examined whether depressive symptoms were associated with differences in arousal ratings depending on the valence of the scenarios. Here, we modeled valence via a quadratic function, given the nonlinear relationship between valence and arousal [52]. This analysis revealed a main effect of valence on arousal (F_1,3341.2_ = 925.587, *p* < 0.001), such that happy scenarios (+1 SD, valence rating of 5.9) and sad scenarios (−1 SD, valence rating of 2.3) were both rated as more arousing than neutral scenarios (mean valence rating of 4.1), (happy—neutral: *β* = 0.403, *t*_1761_ = 15.943, *p* < 0.001, 95% CI = [0.342, 0.464]; sad—neutral: *β* = 0.510, *t*_1478_ = 19.857, *p* < 0.001, 95% CI = [0.449, 0.572]). Sad scenarios were also rated as slightly higher in arousal than happy scenarios (*β* = 0.107, *t*_1025_ = 2.605, *p* = 0.028, 95% CI = [0.009, 0.206]). Additionally, we also observed a main effect of depression (F_1,99.3_ = 20.340, *p* < 0.001), such that individuals reporting higher depressive symptoms experienced the imagined scenarios with greater arousal. Importantly, these main effects were qualified by an interaction of valence and depression (Figure 1c; F_1,3462.1_ = 34.994, *p* < 0.001). Examination of simple slopes revealed that increased depressive symptoms were associated with increased arousal ratings for happy scenarios (*β* = 0.123, *t*_93.9_ = 2.856, *p* = 0.005, 95% CI = [0.038, 0.209]), neutral scenarios (*β* = 0.197, *t*_101.3_ = 4.469, *p* < 0.001, 95% CI = [0.110, 0.284]), and sad scenarios (*β* = 0.128, *t*_95.5_ = 2.941, *p* = 0.004, 95% CI = [0.042, 0.214]). Consistent with ECI, these slopes resulted in diminished differentiation in arousal ratings at high depressive symptoms (+1 SD, main effect of valence level: F_2,2113.21_ = 210.562, *p* < 0.001) compared to low depressive symptoms (−1 SD, main effect of valence level: F_2,2093.6_ = 362.826, *p* < 0.001).

For self-relevance ratings, we observed a main effect of valence on self-relevance (F_1,1987.6_ = 89.170, *p* < 0.001), such that increased valence was associated with higher ratings of self-relevance. We also observed a main effect of depression (F_1,80.6_ = 6.208, *p* = 0.015), such that increased depressive symptoms were associated with higher ratings of self-relevance. As before, these main effects were qualified by a significant interaction of valence and depression (Figure 1d; F_1,3430.1_ = 9.882, *p* = 0.002), with simple slopes indicating that sad and neutral scenarios were rated as more self-relevant as depressive symptoms increased (sad scenarios estimated marginal trend: *β* = 0.143, *t*_99.8_ = 3.283, *p* = 0.001, 95% CI = [0.057, 0.230]; neutral scenarios estimated marginal trend: *β* = 0.103, *t*_81.9_ = 2.471, *p* = 0.016, 95% CI = [0.020, 0.185]), while the self-relevance of happy scenarios did not change (*β* = 0.062, *t*_97.6_ = 1.432, *p* = 0.155, 95% CI = [−0.024, 0.148]). Again, consistent with ECI, these increases in self-relevance ratings led to diminished differentiation between happy, sad, and neutral scenarios at high depressive symptoms (+1 SD, main effect of valence level: F_1,2434.72_ = 51.617, *p* < 0.001) compared to low depressive symptoms (−1 SD, main effect of valence level: F_1,2441.46_ = 88.769, *p* < 0.001).

In sum, analysis of behavioral ratings indicated that increased depression was associated with amplified ratings of valence (more sad and more happy) for emotional scenarios, increased arousal ratings (particularly for neutral scenarios with ambiguous outcomes), and increased self-relevance for neutral and sad imagined events. These effects reflect a general bias in depression for experiencing high arousal, self-relevant emotions—especially for negative and ambiguously neutral information. Importantly, as a result, arousal and self-relevance ratings were less differentiated between sad, neutral, and happy scenarios among participants with high depressive symptoms.

### 3.2. Increased Depression Associates with Less Valence-Related Differentiation of Corrugator Signal to Imagined Events (ECI)

We next examined whether physiological responses were also reflective of similar biases in emotional responding. For these analyses, we focused on EMG corrugator activity during the 16 s when participants read and imagined the scenarios (prior to providing behavioral ratings). We first tested whether corrugator activity successfully tracked the emotionality of the scenarios across all participants. We modeled valence ratings with a quadratic function to accommodate nonlinear relations. As shown in Figure 2, examination of estimated marginal means revealed that valence-related differentiation of the corrugator signal primarily emerged after participants had fully read the scenarios (average read time = 3.7 s, SD = 1.3), and this differentiation was sustained throughout the imagination period. Indeed, when corrugator amplitude was averaged across 4 to 16 s after scenario onset, we observed a significant linear (F_1,87.38_ = 68.051, *p* < 0.001) and nonlinear (F_1,591.24_ = 4.381, *p* = 0.037) effect of valence, indicating decreased corrugator amplitude as valence ratings became more positive.

We then tested whether valence-related differentiation of corrugator amplitude was moderated by individual differences in depression as well as self-relevance ratings, focusing on the imagination period that occurred 4 to 16 s after trial onset. We observed a significant two-way interaction of valence and depression (F_1,3429.7_ = 8.932, *p* = 0.003), which was qualified by a significant three-way interaction of valence, self-relevance, and depression (F_1,3477_ = 5.073, *p* = 0.024). Of note, no significant interactions were found when self-relevance was replaced with arousal ratings (all *p*s > 0.05).

We further unpacked the observed three-way interaction by examining the two-way interaction of depression and valence level (sad, neutral, and happy) at low, mean, and high levels of self-relevance (Figure 3). Significant two-way interactions emerged at low (F_2,3474.08_ = 6.488, *p* = 0.002) and mean (F_2,3481.25_ = 4.649, *p* = 0.01) levels of self-relevance, but not at high self-relevance (F_2,3487.53_ = 0.613, *p* = 0.542). The two-way interactions at low and mean levels of self-relevance were driven by significant differences in the slope of happy scenarios when compared to both sad and neutral scenarios (Table 1). As depressive symptoms increased, corrugator amplitude in response to happy scenarios also increased, while corrugator amplitude in response to sad and neutral scenarios decreased.

The consequences of these trends are made evident when comparing corrugator amplitude between sad, neutral, and happy scenarios at low (−1 SD) and high (+1 SD) levels of depressive symptoms (see Figure 3 for statistical comparisons). For low depressive symptoms, corrugator amplitude was significantly separated between sad, neutral, and happy scenarios (sad > neutral > happy) at all levels of self-relevance, consistent with the sensitivity of this measure to affective valence. For high depressive symptoms, this separation was generally weaker, but primarily driven by a complete loss of differentiation at low self-relevance. Moreover, even at high self-relevance, the dissociation in signal between sad and neutral scenarios was less apparent in highly depressed individuals. Thus, consistent with ECI, increased depression was associated with blunted physiological reactivity to emotional stimuli that contributed to a loss of valence-related differentiation, but primarily for stimuli with low self-relevance.

### 3.3. Corrugator Amplitude Differentiates between the Recall of Highly Self-Relevant Happy or Sad Autobiographical Memories but Is Not Moderated by Depressive Symptoms

Our analysis of corrugator amplitude during the imagination of hypothetical scenarios suggested that the presence of ECI is moderated by self-relevance, such that depression is only associated with blunted physiological reactivity when self-relevance is low, but not when self-relevance is high. We therefore suspected that because autobiographical memories have high self-relevance, corrugator amplitude would not exhibit ECI in depressed individuals while recalling happy or sad memories.

Using the same approach as we did with the EMG data from the imagination task, we first examined the time course of corrugator amplitude throughout the full 20 s that participants were recalling their memories, as a function of whether the memories were happy or sad (Figure 4). Examination of estimated marginal means again revealed that valence-related differentiation of the corrugator signal primarily emerged after participants had read the memory cue and initially retrieved the event (average retrieval time = 2.9 s, SD = 1.5). This differentiation in response was sustained throughout the recall period, as average corrugator amplitude 4 to 20s after memory cue onset was significantly different between recall groups (F_1,79.598_ = 8.791, *p* = 0.004). Interestingly, when we added self-report ratings into the model, we observed a significant interaction with memory arousal (F_1,862.56_ = 6.160, *p* = 0.013), such that valence-related differentiation in corrugator amplitude was enhanced at higher levels of arousal.

We note that similar effects were also observed in self-reported valence. That is, we observed a main effect of recall group (F_1,81.74_ = 2041.447, *p* < 0.001), such that happy memories (estimated marginal mean = 5.93, 95% CI = [5.81, 6.04]) were rated as more happy than sad memories (estimated marginal mean = 2.29, CI = [2.18, 2.40]). This main effect was qualified by a significant interaction with arousal (F_1,1084.83_ = 378.360, *p* < 0.001), such that increased arousal ratings were associated with happier happy memories (*β* = 0.231, *t*_1095_ = 12.087, *p* < 0.001, 95% CI = [0.193, 0.268]) and sadder sad memories (*β* = −0.270, *t*_1090_ = −15.439, *p* < 0.001, 95% CI = [−0.304, −0.235]).

We next tested whether individual differences in depressive symptoms moderated corrugator amplitude during memory recall. We did not observe an interaction of depression with recall group (F_1,79.394_ = 0.135, *p* = 0.714), indicating similar levels of corrugator activity across participants. The lack of an interaction remained even when we focused analyses on the maximal period of separation from 6 to 10 s after memory cue onset (F_1,79.531_ = 0.004, *p* = 0.947), and we observed no interaction of depressive symptoms with arousal ratings in either analysis (both *p*s > 0.05). Finally, although we also tested for differences in self-report ratings in relation to depressive symptoms, we did not find any significant interactions of recall group and depressive symptoms or main effects of depressive symptoms in predicting valence or arousal ratings (all *p*s > 0.05). In sum, we observed similar response profiles across participants when they were ruminating on happy or sad autobiographical memories, irrespective of depressive symptoms.

## 4. Discussion

In the present study, we found evidence that self-relevance is an important moderator for the presence and strength of ECI in individuals with high depressive symptoms, as measured with facial EMG from the corrugator muscle. When reacting to hypothetical emotional events low in self-relevance, participants with higher depressive symptoms exhibited flattened reactivity to both happy and sad scenarios, whereas those with lower depressive symptoms responded with increased corrugator amplitude to sad scenarios and reduced amplitude to happy scenarios. However, when the emotional scenarios were perceived as highly self-relevant, the strength of valence-related differentiation in corrugator amplitude was similar across all levels of depressive symptoms. Likewise, when participants ruminated on autobiographical memories for real-life personal events, we observed similar response profiles of increased corrugator amplitude when recalling sad memories and decreased corrugator amplitude when recalling happy memories, irrespective of individual differences in depressive symptoms. Thus, as the personal relevance of emotional stimuli increased, the presence of ECI as measured with physiological reactivity decreased.

When taken together, the self-report response profile for individuals with high depressive symptoms also generally indicated diminished differentiation in reactivity towards the imagined scenarios. Both arousal and self-relevance ratings became indistinguishable between happy, neutral, and sad scenarios for those with high depressive symptoms, and this lack of differentiation in self-reported emotional experience may have played a role in the blunted physiological reactivity we observed in the EMG data. Only valence ratings were shown to be more differentiated as depressive symptoms increased, although this same heightened differentiation did not map onto corrugator activity, reflecting aberrant response coherence between self-report and physiology among those with high depressive symptoms [30]. That is, individuals with high depressive symptoms appraised scenarios as being more emotional, even though their physiological reactivity did not mirror these effects. Note, however, that these amplified valence ratings seemed to have been specific to the imagination task, given that we observed no relationship of depressive symptoms with self-report valence ratings during autobiographical recall.

The behavioral effects we observed in the imagination task correspond with similar depression-related effects seen in the literature, such as heightened positive reactivity (via self-report) to positive events in EMA studies [11,12,13], mood-congruent negativity biases in attention and memory [16,17,18], responding to uncertain outcomes (neutral scenarios) with greater arousal [53], and increased ratings of personal relevance for negative content [21,54]. When interpreting these effects, it is important to recognize that the stimuli we selected were all written to be self-referential (second-person), even though individual items could vary in their degree of personal relevance. As such, these stimuli may have facilitated amplified ratings of phenomenological characteristics across the board for those with higher depressive symptoms, as depression has been suggested to be associated with greater sensitivity to self-referential content [55].

Regarding the lack of depression-related differences in responding to autobiographical memories, it may also be the case that the novelty of the hypothetical scenarios facilitated greater sensitivity of self-reported affect ratings to the degree of depressive symptoms. In contrast, the autobiographical recall task presented participants with cues that they had self-generated a week prior and thus were already familiar to them. Moreover, the structure of the tasks may have also played a significant role, given that in the autobiographical recall task participants ruminated on only happy or sad memories, and thus by design were less variable in their valence ratings. We may have observed greater variability in emotional reactivity to autobiographical recall if participants were asked to recall multiple types of emotional memories in response to cues that we provided and they had not yet seen. Comparing these different forms of autobiographical recall tasks as they relate to emotional reactivity is a fruitful avenue for future inquiry, given that the role of cue characteristics in shaping emotional response to autobiographical memories remains an active area of research [56]. Nevertheless, our main finding from the autobiographical recall task is that valence-related differentiation in corrugator signal was preserved among those with high depressive symptoms, a finding that complements the corrugator results for highly self-relevant imagined scenarios.

Why does self-relevance moderate the presence of ECI in corrugator physiology? As presented in the introduction, one possibility is that emotional reactivity is not universally diminished in depression but is rather more selective in its expression. This proposal still aligns with evolutionary perspectives of depression that highlight reduced motivated action and a general disengagement from external stimuli [25], but acknowledges that emotional reactivity is retained in depression (if not amplified as some EMA studies show) when the appropriate conditions are met. If affective information is of low personal relevance, then it may feel particularly wasteful to expend effort to engage with that material when in a depressed mood. Instead, depression promotes saving cognitive resources only for material that fits with one’s self-schema. This behavior, however, may perpetuate depressive symptomatology by impeding thoughts of optimistic futures that do not align with one’s current state, even though such optimistic thoughts help to motivate goal-oriented behavior and can help to improve mood [57]. In a related vein, depression is associated with enhanced emotional inertia, such that depressed individuals exhibit greater resistance to emotional change [22]. This resistance is theorized to accompany perseverative thinking styles and ruminative tendencies that maintain a relatively inflexible responsive state [58]. Regarding the present findings, emotional stimuli may therefore need to be high in self-relevance to overcome this inert response profile.

The findings presented here offer several methodological implications for future researchers in the field to consider. First and foremost, self-relevance ratings should be measured and incorporated into analyses, as we have shown that such appraisals significantly shape psychophysiological responses to emotional stimuli in depression. We also suggest using tasks that are self-referential by design. Our use of mental imagery, for instance, presented participants with novel episodic events that they could experience in real life. We were therefore able to conduct a laboratory study with timed experimental tasks and physiological recording while still engaging participants with ecologically valid stimuli. This approach may better translate to real-life encounters, for which EMA studies have observed findings that diverge from the ECI model [11,12,13]. Our results also highlight the utility of using facial EMG to dissociate emotional reactivity based on valence. Taken together, then, our findings demonstrate the importance of which methods researchers use to measure emotion (self-report vs. physiological recording) and shed light on factors that significantly moderate emotional reactivity, including individual differences in depressive symptoms and features of the emotional stimulus.

Nevertheless, several limitations of the study should be noted. First, we measured number of depressive symptoms with the CES-D questionnaire, which has been shown to have acceptable depression screening accuracy when using cut-off scores ranging from 16–20 [51], although we did not verify a depression diagnosis and therefore cannot make conclusive claims regarding the presence of ECI in major depressive disorder. Nevertheless, as dimensional perspectives gain traction in clinical research [59], it is imperative to test if models of depressive behavior (such as ECI) scale with the number of depressive symptoms even in non-clinical samples, which we have shown here. Second, our sample consisted of young adults, and whether similar ECI effects emerge in older adults remains unclear, although this would be an interesting avenue for future research to explore given differences in emotional reactivity between older and younger adults [60]. Third, although all participants were shown the same scenario texts, we had no control over how participants created their imaginative scenes. Increased depression has been shown to associate with impairments in mental imagery, such as reduced vividness for positive mental images [61], which may have contributed to the effects we observed. Indeed, it may be the case that emotional scenarios with low self-relevance were more difficult for depressed individuals to imagine, and this difference in mental imagery ability contributed to the diminished physiological responding. Future research should also collect ratings of vividness and ease of imagination to test this possibility. Finally, as mentioned, our autobiographical recall task consisted of ruminating on just happy or sad memory cues provided by participants, although different effects may have been observed if we had created the cues for participants, if they had recalled happy and sad memories in alteration, or if we allowed participants to freely ruminate on whatever content first came to mind. Yet, regardless of these alternative approaches, it remains compelling that we did not observe a relationship between depressive symptoms and corrugator response as participants ruminated on autobiographical memories, perhaps due to their strong personal relevance.

In conclusion, here we have shown that increased depressive symptoms are associated with a loss of emotional differentiation in self-report ratings as well as facial EMG from the corrugator muscle. Importantly, though, the latter finding was moderated by individual differences in self-relevance, such that imagined scenarios perceived as highly self-relevant and autobiographical memories—which are personally relevant by definition—did not elicit the same pattern of blunted reactivity. We therefore propose that self-relevance is a key determinant of ECI, such that emotional reactivity in depression is shaped by more selective processing of self-relevant affective information.

## Figures and Tables

**Figure 1 brainsci-13-00843-f001:**
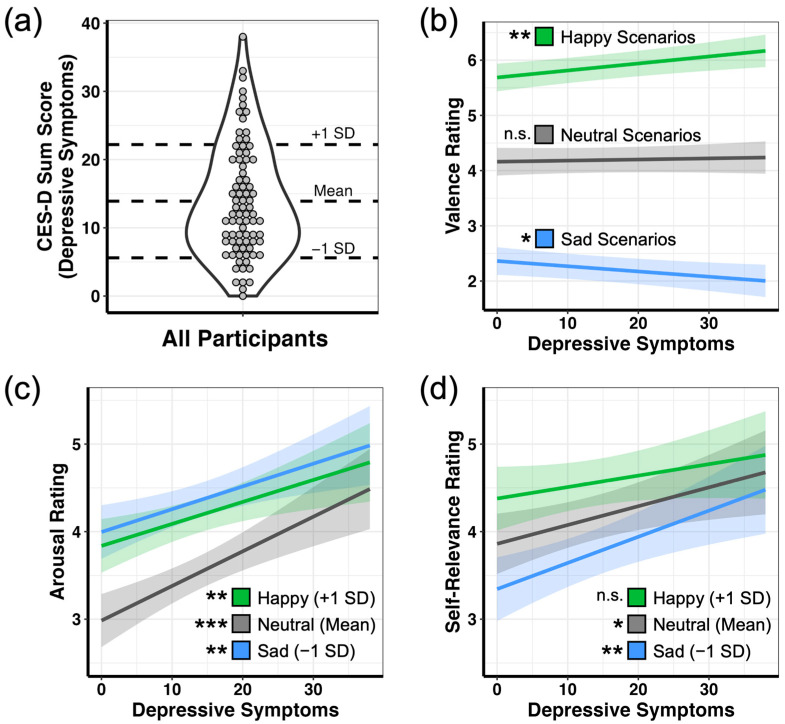
Depression shapes the phenomenological experience of imagined emotional scenarios: (**a**) distribution of depressive symptoms across all participants; (**b**) relationship of depressive symptoms with valence ratings for scenarios grouped by normative categories; (**c**) relationship of depressive symptoms with arousal ratings, shown at three levels of valence ratings—happy (+1 SD, 5.9), neutral (mean, 4.1), and sad (−1 SD, 2.3); (**d**) relationship of depressive symptoms with self-relevance ratings, shown at the same three levels of valence ratings. Plots b-d depict estimated marginal means. Shaded areas represent 95% confidence intervals. For all analyses, n = 81. *** *p* < 0.001; ** *p* < 0.01; * *p* < 0.05; n.s. not significant.

**Figure 2 brainsci-13-00843-f002:**
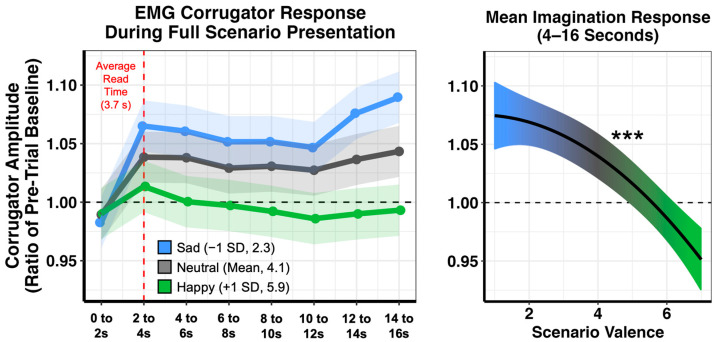
Corrugator amplitude (compared to pre-trial baseline) for all participants during the full 16 s that each scenario is presented on the screen. Sad and neutral scenarios increased in amplitude, whereas happy scenarios generally remained at baseline. Emotion-related differentiation of the corrugator signal primarily emerged after the reading period. Shown on the right is the corrugator response averaged across 4 to 16 s (the imagination period) for the full range of scenario valence. Plots depict estimated marginal means. Shaded areas represent 95% confidence intervals. For all analyses, n = 81. *** *p* < 0.001.

**Figure 3 brainsci-13-00843-f003:**
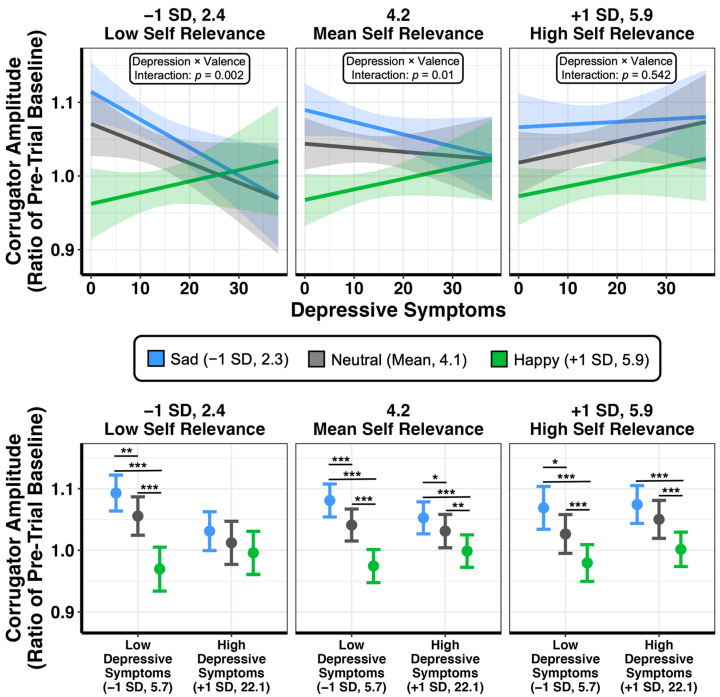
The moderating effect of depression and self-relevance on the relationship between scenario valence and corrugator amplitude during the 4 to 16 s after scenario onset (the imagination period). The top plots show the relationship of depressive symptoms with estimated marginal means of corrugator amplitude, separately for each level of scenario valence and self-relevance. The interaction of depression and valence is diminished as self-relevance increases. The bottom plots depict comparisons between happy, neutral, and sad scenarios for low and high depressive symptoms. At each level of self-relevance, comparisons are Bonferroni-corrected within depression level. For individuals with high depressive symptoms, differences in corrugator amplitude are absent when self-relevance is low. Plots depict estimated marginal means. Shaded areas and error bars represent 95% confidence intervals. For all analyses, n = 81. *** *p* < 0.001, ** *p* < 0.01, * *p* < 0.05.

**Figure 4 brainsci-13-00843-f004:**
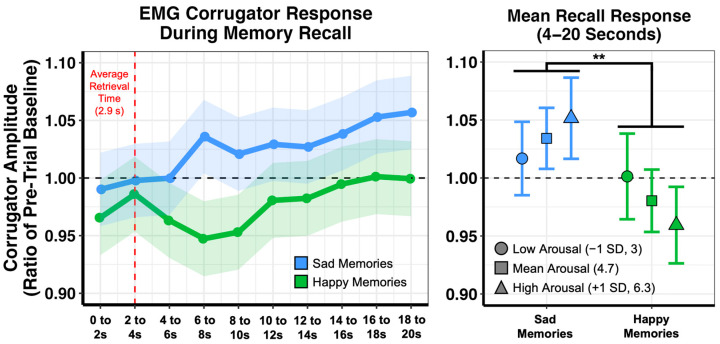
Corrugator amplitude (compared to pre-trial baseline) for all participants during the full 20 s that the memory cue was presented on the screen during the autobiographical recall task. Recalling sad memories (blue) resulted in an increase in amplitude, whereas recalling happy memories (green) resulted in a decrease in amplitude. Emotion-related differentiation of the corrugator signal primarily emerged after participants indicated they had retrieved the memory. Shown on the right is the corrugator response averaged across 4 to 20 s (the recall/rumination period after initial retrieval of the event), which exhibited a significant difference between groups. This difference in corrugator amplitude was most apparent for highly arousing memories. Plots depict estimated marginal means. Shaded areas and error bars represent 95% confidence intervals. For all analyses, n = 80 (39 participants recalled happy memories and 41 participants recalled sad memories). ** *p* < 0.01.

**Table 1 brainsci-13-00843-t001:** The moderating effect of depression and self-relevance on the relationship between scenario valence and corrugator amplitude.

Moderator Level	Depression Slope	*t*	95% CI
Low Self-Relevance (−1 SD)			
Sad (−1 SD)	−0.118 **^A^**	−2.848 **	−0.200, −0.037
Neutral (Mean)	−0.083 **^A^**	−1.792	−0.174, 0.008
Happy (+1 SD)	0.050 **^B^**	1.057	−0.043, 0.144
Mean Self-Relevance			
Sad (−1 SD)	−0.054 **^A^**	−1.560	−0.122, 0.014
Neutral (Mean)	−0.019 **^A^**	−0.526	−0.090, 0.052
Happy (+1 SD)	0.046 **^B^**	1.313	−0.023, 0.116
High Self-Relevance (+1 SD)			
Sad (−1 SD)	0.010 **^A^**	0.249	−0.072, 0.093
Neutral (Mean)	0.045 **^A^**	1.103	−0.036, 0.126
Happy (+1 SD)	0.042 **^A^**	1.139	−0.031, 0.115

Note. Different letters after the slope values indicate differences in slope at *p* < 0.05 (Bonferroni-corrected). All variables were standardized prior to regression. ** *p* < 0.01.

## Data Availability

The data presented in this study are available on request from the corresponding author.

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
