# Peer review of "Self-Relevance Moderates the Relationship between Depressive Symptoms and Corrugator Activity during the Imagination of Personal Episodic Events"

_brainsci, 2023, doi:10.3390/brainsci13060843_

Round 1

Reviewer 1 Report

The manuscript presents a study that investigates the link between depression and emotion context insensitivity (ECI), which is a reduced emotional responsiveness to positive and negative stimuli. The authors suggest that self-relevance may moderate the presence of ECI, and the study involves measuring self-report and facial electromyography (EMG) responses from the corrugator muscle in relation to scenarios with varying degrees of self-relevance. The results indicate that individuals with high depressive symptoms exhibit reduced emotional reactivity to scenarios with low self-relevance while remaining sensitive to valence for highly self-relevant scenarios. This suggests that ECI may be primarily associated with reduced emotional reactivity towards material that is perceived as low in self-relevance. The results are clearly presented, and the conclusion is supported by the data.

Suggestions for revision:

1. Please include the sample size (n) in the figure captions. It is also recommended to indicate the exact p-value and R-value in the figures.

2. It would be helpful to cite Figure 1(b, c, d) in the paragraph for easier understanding.

3. Please ensure that references are formatted correctly.

Author Response

Reviewer 1: The manuscript presents a study that investigates the link between depression and emotion context insensitivity (ECI), which is a reduced emotional responsiveness to positive and negative stimuli. The authors suggest that self-relevance may moderate the presence of ECI, and the study involves measuring self-report and facial electromyography (EMG) responses from the corrugator muscle in relation to scenarios with varying degrees of self-relevance. The results indicate that individuals with high depressive symptoms exhibit reduced emotional reactivity to scenarios with low self-relevance while remaining sensitive to valence for highly self-relevant scenarios. This suggests that ECI may be primarily associated with reduced emotional reactivity towards material that is perceived as low in self-relevance. The results are clearly presented, and the conclusion is supported by the data.

Response: We thank the reviewer for these positive remarks and suggestions for how to improve the manuscript.

Suggestions for revision:

  1. Please include the sample size (n) in the figure captions. It is also recommended to indicate the exact p-value and R-value in the figures.

Response: We thank the reviewer for this suggestion. The sample size is now provided in all figure captions. We refer readers to the text immediately preceding each figure for comprehensive reporting of all statistical values (F-stats, t-stats, beta values, p-values, and confidence intervals). For conciseness, we use standard notation in all figures to indicate effects that are at p < .05 (*), p < .01 (**), or p < .001 (***).  

  1. It would be helpful to cite Figure 1(b, c, d) in the paragraph for easier understanding.

Response: We thank the reviewer for this suggestion. We now cite each of the plots (1a, 1b, 1c, and 1d) separately in the text.

  1. Please ensure that references are formatted correctly.

Response: We have formatted references in accordance with the MDPI instructions for authors.

Reviewer 2 Report

It is a strength of the study that the relationship between depression symptoms, emotion context insensitivity (ECI), and self-relevance was examined using both self-report measures and facial electromyography (EMG). This allows for a more thorough understanding of the mechanisms underlying the observed effects. I will call it pilot study. 

Comments to address 

The study's relatively small sample size (n = 81) is one potential drawback. A bigger sample size might improve the generalizability of the results, I would like to see power analysis.

A self-reported measure of depression symptoms. However, the manuscript mixes depression and depressive symptoms. 

The authors' intriguing and potentially significant contribution to the literature is that self-relevance may modulate the link between depression and ECI. Their theory is supported by the observation that those with high levels of depressive symptoms had decreased corrugator response for sad, neutral, and cheerful scenarios with low self-relevance, but not for highly self-relevant scenarios. This raises the possibility that ECI may be more pronounced in content with a low degree of self-relevance, which has significant therapeutic interventional implications. Mechanism need better discussion and what are +/-. 

Give the qualifying requirements, as well as the sources and procedures used to choose the participants/population. Did you screen for specific issues?

All outcomes, exposures, predictors, potential confounders, and effect modifiers should be precisely defined. Describe the diagnostic standards, if any e.g. cases self-identified as depression using DSM

Describe the methods used to handle quantitative variables in the analyses. Describe the classifications that were chosen, if relevant, and why.

Author Response

Reviewer 2: It is a strength of the study that the relationship between depression symptoms, emotion context insensitivity (ECI), and self-relevance was examined using both self-report measures and facial electromyography (EMG). This allows for a more thorough understanding of the mechanisms underlying the observed effects. I will call it pilot study.

Response: We thank the reviewer for these positive remarks and suggestions for how to improve the manuscript.

Comments to address

The study's relatively small sample size (n = 81) is one potential drawback. A bigger sample size might improve the generalizability of the results, I would like to see power analysis.

Response: We thank the reviewer for this comment. As noted in the manuscript, the present investigation is part of a larger multi-session study on mood-congruent memory, for which we recruited more than 80 individuals to have sufficient power to shift mood states with the autobiographical memory task. As such, we did not perform a prospective power analysis for these secondary analyses on emotion context insensitivity (ECI) as they relate to CES-D scores. We also note that post-hoc power analyses have been shown to be logically invalid and practically misleading and therefore are not recommended to be performed [1]. Nevertheless, our sample size is similar to other studies in the field that have also examined ECI with the CES-D or BDI scales [2–5]. Moreover, our analytical approach is more sensitive to detecting ECI effects by always evaluating CES-D scores on a continuous scale, which has been shown to be more appropriate than merely dichotomizing the continuous variable into high and low levels [6,7]. Furthermore, our use of linear mixed effects regression accounts for the random effects of both subjects and stimuli, which vastly improves the power of our analyses and the generalizability of the observed effects compared to conventional statistical techniques that do not respect this random effects structure [8].

  1. Dziak, J.J.; Dierker, L.C.; Abar, B. The Interpretation of Statistical Power after the Data Have Been Gathered. Curr. Psychol. 2020, 39, doi:10.1007/s12144-018-0018-1.
  2. Panaite, V.; Koval, P.; Dejonckheere, E.; Kuppens, P. Emotion Regulation and Mood Brightening in Daily Life Vary with Depressive Symptom Levels. Cogn. Emot. 2019, 33, doi:10.1080/02699931.2018.1543180.
  3. Brinkmann, K.; Schüpbach, L.; Joye, I.A.; Gendolla, G.H.E. Anhedonia and Effort Mobilization in Dysphoria: Reduced Cardiovascular Response to Reward and Punishment. Int. J. Psychophysiol. 2009, 74, doi:10.1016/j.ijpsycho.2009.09.009.
  4. Thibodeau, R. Individual Differences in Depressive Symptoms Are Associated with Impaired Incentive, but Not Aversive Motivation. Pers. Individ. Dif. 2011, 50, doi:10.1016/j.paid.2010.10.025.
  5. Ellis, A.J.; Beevers, C.G.; Wells, T.T. Emotional Dysregulation in Dysphoria: Support for Emotion Context Insensitivity in Response to Performance-Based Feedback. J. Behav. Ther. Exp. Psychiatry 2009, 40, doi:10.1016/j.jbtep.2009.05.002.
  6. MacCallum, R.C.; Zhang, S.; Preacher, K.J.; Rucker, D.D. On the Practice of Dichotomization of Quantitative Variables. Psychol. Methods 2002, 7, doi:10.1037/1082-989X.7.1.19.
  7. Fernandes, A.; Malaquias, C.; Figueiredo, D.; da Rocha, E.; Lins, R. Why Quantitative Variables Should Not Be Recoded as Categorical. J. Appl. Math. Phys. 2019, 07, doi:10.4236/jamp.2019.77103.
  8. DeBruine, L.M.; Barr, D.J. Understanding Mixed-Effects Models Through Data Simulation. Adv. Methods Pract. Psychol. Sci. 2021, 4, doi:10.1177/2515245920965119.

A self-reported measure of depression symptoms. However, the manuscript mixes depression and depressive symptoms.

Response: We thank the reviewer for noting this choice in language. In the current study we used the Center for Epidemiologic Studies Depression Scale (CES-D), which is a well-validated tool for measuring symptoms associated with depression. As such, higher scores on the CES-D (i.e., higher depressive symptoms) are consistently referred to in the literature as indicative of increased depression, hence why we used these terms interchangeably. Importantly, all analyses used the CES-D as a continuous measure, such that the observed effects reflect behavioral and psychophysiological responses that are associated with higher depressive symptoms/increased depression. However, given that we did not specifically test for a diagnosis of major depressive disorder, we are careful to never refer to these effects as reflecting differences between clinical and non-clinical groups. We note this limitation on page 15:

“We measured number of depressive symptoms with the CES-D questionnaire, which has been shown to have acceptable depression screening accuracy when using cut-off scores ranging from 16-20 [51], although we did not verify a depression diagnosis and therefore cannot make conclusive claims regarding the presence of ECI in major depressive disorder. Nevertheless, as dimensional perspectives gain traction in clinical research [59], it is imperative to test if models of depressive behavior (such as ECI) scale with the number of depressive symptoms even in non-clinical samples, which we have shown here.”

The authors' intriguing and potentially significant contribution to the literature is that self-relevance may modulate the link between depression and ECI. Their theory is supported by the observation that those with high levels of depressive symptoms had decreased corrugator response for sad, neutral, and cheerful scenarios with low self-relevance, but not for highly self-relevant scenarios. This raises the possibility that ECI may be more pronounced in content with a low degree of self-relevance, which has significant therapeutic interventional implications. Mechanism need better discussion and what are +/-.

Response: We thank the reviewer for these positive remarks. Regarding discussion of the mechanism, we refer the reviewer to page 14 of the manuscript, where we have expanded our discussion on the role of self-relevance in shaping ECI:

“Why does self-relevance moderate the presence of ECI in corrugator physiology? As presented in the introduction, one possibility is that emotional reactivity is not universally diminished in depression but is rather more selective in its expression. This proposal still aligns with evolutionary perspectives of depression that highlight reduced motivated action and a general disengagement from external stimuli [25], but acknowledges that emotional reactivity is retained in depression (if not amplified as some EMA studies show) when the appropriate conditions are met. If affective information is of low personal relevance, then it may feel particularly wasteful to expend effort to engage with that material when in a depressed mood. Instead, depression promotes saving cognitive resources only for material that fits with one’s self-schema. This behavior, however, may perpetuate depressive symptomatology by impeding thoughts of optimistic futures that do not align with one’s current state, even though such optimistic thoughts help to motivate goal-oriented behavior and can help to improve mood [57]. In a related vein, depression is associated with enhanced emotional inertia, such that depressed individuals exhibit greater resistance to emotional change [22]. This resistance is theorized to accompany perseverative thinking styles and ruminative tendencies that maintain a relatively inflexible responsive state [58]. Regarding the present findings, emotional stimuli may therefore need to be high in self-relevance to overcome this inert response profile.”

Give the qualifying requirements, as well as the sources and procedures used to choose the participants/population. Did you screen for specific issues?

Response: We thank the reviewer for this question. As we now note on line 165, “Inclusion criteria included an age range of 18-39 years old, as well as no history of a neurological condition or psychiatric disorder (although we did not specifically test for the presence or absence of a disorder).”

All outcomes, exposures, predictors, potential confounders, and effect modifiers should be precisely defined. Describe the diagnostic standards, if any e.g. cases self-identified as depression using DSM

Response: We did not apply specific diagnostic standards, as we only used scores on the CES-D questionnaire as a measure of depressive symptoms. All variables are described in the methods and results sections, including valence (line 259), arousal (lines 259-260), self-relevance (lines 260-261), and CES-D (lines 362-369, and now also lines 188-191).

Describe the methods used to handle quantitative variables in the analyses. Describe the classifications that were chosen, if relevant, and why.

Response: All quantitative variables are treated as continuous variables in analyses. Simple effects/slopes are evaluated using estimated marginal means after testing the full model with the appropriate variables. These effects are examined at -1 SD below the mean, at the mean, and at +1 SD above the mean, but nonetheless reflect continuous (not discrete) effects. The only discrete variable is the group assignment to recalling sad or happy autobiographical memories (Figure 4), as indicated on line 354. 

Round 2

Reviewer 2 Report

Thank you for addressing my comments.